# VmsR, a LuxR-Type Regulator, Contributes to Virulence, Cell Motility, Extracellular Polysaccharide Production and Biofilm Formation in *Xanthomonas oryzae* pv. *oryzicola*

**DOI:** 10.3390/ijms25147595

**Published:** 2024-07-11

**Authors:** Yaqi Zhang, Xiyao Zhao, Jiuxiang Wang, Lindong Liao, Huajun Qin, Rongbo Zhang, Changyu Li, Yongqiang He, Sheng Huang

**Affiliations:** 1State Key Laboratory for Conservation and Utilization of Subtropical Agro-Bioresources, College of Life Science and Technology, Guangxi University, Nanning 530004, China; 2108401004@st.gxu.edu.cn (Y.Z.); 2108301084@st.gxu.edu.cn (X.Z.); 2008401006@st.gxu.edu.cn (J.W.); 2108391025@st.gxu.edu.cn (L.L.); hjqin@st.gxu.edu.cn (H.Q.); 2208401004@st.gxu.edu.cn (R.Z.); 2308391015@st.gxu.edu.cn (C.L.); 2College of Agronomy, Guangxi University, Nanning 530004, China

**Keywords:** *Xanthomonas*, LuxR-type regulator, virulence, motility, extracellular polysaccharides, biofilm

## Abstract

LuxR-type regulators play pivotal roles in regulating numerous bacterial processes, including bacterial motility and virulence, thereby exerting a significant influence on bacterial behavior and pathogenicity. *Xanthomonas oryzae* pv. *oryzicola*, a rice pathogen, causes bacterial leaf streak. Our research has identified VmsR, which is a response regulator of the two-component system (TCS) that belongs to the LuxR family. These findings of the experiment reveal that VmsR plays a crucial role in regulating pathogenicity, motility, biofilm formation, and the production of extracellular polysaccharides (EPSs) in *Xoc* GX01. Notably, our study shows that the *vmsR* mutant exhibits a reduced swimming motility but an enhanced swarming motility. Furthermore, this mutant displays decreased virulence while significantly increasing EPS production and biofilm formation. We have uncovered that VmsR directly interacts with the promoter regions of *fliC* and *fliS*, promoting their expression. In contrast, VmsR specifically binds to the promoter of *gumB*, resulting in its downregulation. These findings indicate that the knockout of *vmsR* has profound effects on virulence, motility, biofilm formation, and EPS production in *Xoc* GX01, providing insights into the intricate regulatory network of *Xoc*.

## 1. Introduction

Bacterial leaf streak (BLS) is one of the most important diseases of rice in Asia, Australia, and Africa. *Xanthomonas oryzae* pv. *oryzicola* (*Xoc*) is a bacterial pathogen responsible for BLS, which causes significant yield losses in rice fields, with estimates ranging from 10% to 30% in severely infected areas. This translates into substantial economic losses for farmers and the rice industry globally [1,2]. *Xoc* is a Gram-negative bacterium and possesses a single polar flagellum, which plays important roles in bacterial motility and pathogenicity [3,4]. The virulence mechanisms of *Xoc* are greatly complicated. There is little research on virulence mechanisms, and the control measures for BLS are poorly developed. Bacterial plant pathogens possess a range of pathogenicity factors, including type Ⅲ effector, extracellular polysaccharides (EPSs), lipopolysaccharides, extracellular enzymes, motility, and biofilm [5,6,7]. These pathogenicity factors facilitate the infection of host tissues by the pathogen. The regulation of these pathogenicity factors is mediated by intricate regulatory mechanisms, including a two-component system (TCS). TCS plays a pivotal role in orchestrating the bacterial response to environmental cues and fine tuning the expression of virulence-associated genes, thereby ensuring the precise control of pathogenicity.

LuxR-family transcriptional regulators are generally 250 amino acids in size with a helix–turn–helix LuxR (HTH-LuxR) domain at the C-terminus and a variable domain at the N-terminus that interacts with signaling substances, for example, the quorum sensing (QS) system, TCSs, and other signals [8,9,10]. The HTH-LuxR domain has the capacity to bind with a distinct gene promoter, thereby either stimulating or suppressing the expression of the intended genes. It is widely known that LuxR plays a role in the QS system. The *N*-acyl homoserine lactones (AHLs) are small, diffusible molecules used as communication signals in a large variety of proteobacteria [11,12]. A typical AHLs-mediated QS system is composed of a LuxR protein and a cognate LuxI protein, which were first identified in *Vibrio fischeri* [13,14]. This kind of LuxR-type protein has an N-terminal AHLs-binding domain and a C-terminal HTH-LuxR DNA-binding domain.

TCSs, comprising a membrane-bound histidine kinase sensor (HKs) and a cytoplasmic response regulator (RR), represent the primary molecular mechanisms utilized by prokaryotes to perceive and react to environmental cues [15]. TCSs play an important role in the pathogenicity of microbes. In 1986, Ninfa et al. revealed in *Escherichia coli* (*E. coli*) a two-component signal transduction system, wherein the nitrogen-regulatory protein NRI is intricately regulated by the histidine kinase NRII, constituting a receptor–regulator pair that governs nitrogen metabolism. This marked the initial recognition of such a system by researchers [16]. In *Xanthomonads* spp., the RpfC/RpfG, responsible for QS signal transduction, plays a role in positively regulating pathogenicity factors [17,18]. In *Xanthomonas campestris* pv. *vesicatoria*, the function of HrpG was determined to occupy a pivotal position at the apex of the *hrp* gene regulatory cascade. The amino acid sequence of HrpG exhibits similarity to RR proteins belonging to the OmpR-family subclass of TCSs [19]. By interacting with the downstream AraC-family transcription factor HrpX, HrpG regulates the expression of the type Ⅲ secretion system (T3SS), which is a crucial transportation machinery responsible for delivering effectors into host cells [20].

Bacteria display diverse locomotor patterns based on whether they reside in a liquid medium or are in contact with a solid substrate [21,22]. In liquid environments, certain bacteria produce flagella and navigate in three dimensions as individual swimmers; however, upon encountering a solid surface, they initiate swift swarming locomotion in two dimensions across the substrate [23,24]. Moreover, swimmer and swarmer cells exhibit physiological differences, as the transition to surfaces involves a period of immobility known as a lag phase, which could serve as a window for cellular differentiation triggered by surface contact [25]. Swarming motility was first recorded for *Proteus* species and subsequently observed to be widespread among flagellated bacteria [26]. Unlike swimming, bacteria exhibit a lag before moving. Bacteria must reach higher cell densities, and they generally require energy-rich media for swarming to begin [25]. Flagella are the primary organelles responsible for bacterial motility in aqueous environments. Flagella are essential multi-functional structures composed primarily of a 39 kDa flagellin protein (FliC) and a 56 kDa flagellar cap protein (FliD). FliS acts as a flagellin protein (FliC) export chaperone. When the expression of these flagellar proteins is reduced, the bacteria are unable to assemble functional flagella, leading to a decrease in motility. In natural habitats, bacteria often compete with other microorganisms for resources and space. Decreased flagellar stability due to reduced gene expression can make bacteria more susceptible to these stresses, potentially reducing their survival rates in adverse conditions. For pathogenic bacteria, flagella play a crucial role in virulence, as they are involved in host cell attachment, invasion, and biofilm formation. This impairment can affect the ability of bacteria to colonize surfaces, move toward nutrients, or escape from harmful environments [27,28]. In many bacteria, mutations that overcome these requirements map to TCSs or to regulators that increase flagella synthesis.

The formation of biofilms involves several phases: initial attachment, the establishment of microcolonies and macrocolonies, and ultimately, detachment or disassembly [29,30]. EPSs play crucial roles throughout each of these stages: facilitating adhesion to surfaces, facilitating the formation of intricate structures by enhancing microbial interactions, and facilitating the release of these interactions to promote the dissolution of the biofilm. Bacteria are capable of producing multiple EPSs, which are significant in diverse strains and varying environmental conditions, encompassing surface substrate, nutritional availability, and flow rate [31].

These pathogenicity factors play a vital role in determining the virulence of *Xoc*, as they contribute significantly to its ability to cause disease. Typically, these pathogenicity factors are intricately regulated by the TCSs. Conducting a thorough study on TCSs and pathogenicity factors is highly imperative, as it offers crucial insights into the mechanisms underlying disease causation and virulence of *Xoc*.

Despite the crucial role TCSs play in regulating bacterial gene expression and cellular activities, their investigation in *Xanthomonas* spp. remains limited, necessitating further exploration to fully understand their functional significance within this genus. In this study, we identified the XOC_2507 regulator, a member of the LuxR family within the TCSs, in *Xoc* GX01 [32]. Our objective was to elucidate the role of XOC_2507 in virulence, motility, EPS production, and biofilm formation in *Xoc* GX01 through gene knockdown experiments. Additionally, we aimed to explore the underlying mechanisms regulated by XOC_2507 expression using high-throughput transcriptional sequencing. RNA sequencing (RNA-Seq) analysis revealed that numerous genes involved in diverse functions were significantly differentially expressed in the ∆*XOC_2507* strain. Moreover, XOC_2507 can bind to the promoters of *fliC*, *fliS* and *gumB* in vitro. Herein, XOC_2507 is designated as VmsR (Virulence, Motility and Extracellular polysaccharides and Regulator). This study identifies a novel regulator, VmsR, which regulates the expression of virulence-associated flagella and EPS in *Xoc*.

## 2. Results

### 2.1. Identification of VmsR, a Putative LuxR-Type Regulator of Xoc, and Generation of vmsR Mutant and Its Complementary Strain

An open-reading frame (ORF) encoding 210 amino acids residues was identified in the genome sequence of the *Xoc* GX01 (GenBank: CP043403.1) [32], located from nucleotide 2453070 to 2453702, with the locus tag of *XOC_2507*. In this study, the gene has been designated as *vmsR*, exhibiting a characteristic architecture that is typical of the LuxR family of response regulators. Specifically, it comprises a receiver domain situated at the N-terminus, responsible for signal reception, and a helix–turn–helix (HTH) domain positioned at the C-terminus, enabling DNA binding (Figure 1A). The VmsR gene and the quorum-sensing signal receptor XocR exhibit no discernible similarities at the nucleotide or protein level (Appendix A). XocR possesses an Autoind_bind domain at the N-terminus and HTH-LuxR domain at the C-terminus (Figure 1B). The *xocR* is located from nucleotide 1409638 to 1410402 with the locus tag of *XOC1422* in the genome of *Xoc* BLS256 (GenBank: CP003057.2) [33]. The Autoind_bind domain binds AHLs, which are also known as autoinducers. The upstream of VmsR encompasses a gene cluster that is integral to the synthesis of flagellum, which is a vital component for bacterial motility and pathogenesis (Figure 1C). To investigate the functional role of the VmsR regulator, the ∆*vmsR* strain was constructed and analyzed (Appendix A).

To determine the evolutionary relationship between VmsR and diverse *Xanthomonads* spp. LuxRs, a phylogenetic tree was constructed with MEGA11 software. We selected 22 LuxR-type regulatory proteins from diverse *Xanthomonads* species. Upon analysis, the 22 genes were systematically classified into two distinct categories. Taking VmsR of *Xoc* GX01 as the representative protein, it is classified into one category, while XocR of *Xoc* BLS256, as the representative protein, is grouped into a separate category (Figure 1D). VmsR shares very close evolutionary relationships with the LuxR-type HTH regulators XOC_2369 of *Xoc* BLS256 (Figure 1D). In addition, XocR has relatively close relationships with Xoc_3513 of *Xoc* GX01 (Figure 1D). This suggests that the VmsR in *Xoc* GX01 is also a LuxR-type transcriptional regulator because it is an *Xoc* regulator that is different from XocR in the cognate strain *Xoc* BLS256.

### 2.2. VmsR Positively Affects the Virulence of Xoc GX01

To determine whether VmsR contributes to the virulence of *Xoc* GX01, the virulence of *Xoc* strains was tested in this work. The pathogenicity ability of wild-type strain GX01, mutant strain ∆*vmsR*, and its complementary strain C∆*vmsR* was evaluated by inoculation onto the leaf of *Oryza sativa* L. ssp. *japonica* cultivar *Nipponbare.* Fifteen days after inoculation, ∆*vmsR* caused disease symptoms with a lesion length of 16.8 ± 1.64 mm, which was significantly less severe than that caused by the GX01 (lesion length 32.53 ± 2.16 mm). Notably, the C∆*vmsR* strain exhibited virulence symptoms (lesion length 30.47 ± 1.82 mm) similar to those of the wild-type strain GX01 (Figure 2A,B). This result indicates that C∆*vmsR* could restore the impaired virulence of ∆*vmsR*. These results suggest that VmsR positively regulates the pathogenic capacity demonstrated by *Xoc* GX01.

### 2.3. VmsR Is an Important Regulator Involved in Various Cellular Processes of Xoc

To determine whether VmsR plays a crucial role in virulence, we compared the ∆*vmsR* with its wild-type strain *Xoc* GX01 through RNA-Seq. For all six samples analyzed (3 × ∆*vmsR* and 3 × GX01), more than 80% of the total reads were successfully mapped back to the assembled transcriptome. From the total of 5006 genes obtained in the transcriptome of *Xoc* GX01, 140 genes were found to be statistically significant differentially expressed genes (DEGs) between ∆*vmsR* and GX01 using a more stringent threshold (FDR ≤ 0.05 and |Fold Change| ≥ 2). Among these genes, 76 genes were upregulated and 64 genes were downregulated in ∆*vmsR* (Figure 3A).

To gain a deeper understanding of the functionalities of genes regulated by VmsR, we performed a functional grouping analysis based on the genomic annotations of the *Xoc* GX01 [34,35]. Based on the clusters of orthologous groups (COGs), out of the 140 DEGs in the ∆*vmsR* strain, 104 were assigned to 15 various functional categories and 36 were predicted to encode hypothetical proteins or proteins that have not been given a functional category (Figure 3B). The primary functional groupings are “cellular processes” and “pathogenicity and adaption”. In total, 18 and 11 genes fell into these two categories, respectively (Figure 3B). Notably, 105 genes were assigned to “cell envelope and cell structure” (10) and “translation” (12). Appendix A shows a detailed categorization of DEGs.

Consistent with the finding that LuxR regulates cell motility [36,37,38], the identified transcriptional profiles reveal that VmsR has a crucial impact on a number of genes that contribute to cell motility. Upon a thorough examination of the functional classification of the 140 DEGs identified in the ∆*vmsR* mutant of *Xoc* GX01, it is evident that the pathways most significantly affected are those associated with cellular processes, which are fundamental to the survival, growth, and virulence of the bacterium. This observation underscores the central role that VmsR plays in regulating vital cellular activities, particularly those related to motility. Consequently, there is a pressing need to delve deeper into the regulatory mechanisms employed by VmsR in this regard. The pertinent gene clusters that are implicated in these processes are outlined as follows: The four flagellar-related DEGs consist of three genes, namely *XOC_2502*/*fliC*, *XOC_2503*/*fliD*, and *XOC_2504*/*fliS*, which encode proteins integral to the formation of flagellar filaments. Additionally, there is a gene encoding the upstream regulator of flagella, *flrA*. Fourteen of the DEGs encode methyl-accepting chemotaxis proteins: *XOC_2212*, *XOC_2478*, *XOC_2590*, *XOC_2602*, *XOC_2603*, *XOC_2604*, *XOC_2605*, *XOC_2606*, *XOC_2610*, *XOC_2612*, *XOC_2614*, *XOC_2615*, *XOC_2617*, and *XOC_2861*. The expression levels of the mRNA for these genes corresponded closely with the RNA-Seq data (Figure 3C). The findings demonstrate a significant role of VmsR in influencing the motility of *Xoc*. Taken together, the results from the transcriptome analysis reveal that VmsR acts as an essential regulator involved in many cellular processes in *Xoc*.

### 2.4. Loss of vmsR in Xoc Enhanced the Swarming Motility but Impaired the Swimming Motility

Based on the RNA-Seq analysis, we hypothesize that VmsR is linked to bacterial motility. To validate this prediction, we conducted a study to measure the swarming and swimming motilities of *Xoc* GX01, the ∆*vmsR* mutant strain, and the complementary strain C∆*vmsR*. To assess swarming motility, *Xoc* strains were inoculated in a semi-solid NA medium containing 0.6% agar and incubated at 28 °C for 3 days. The wild-type strain GX01 exhibited an average bacterial colony diameter of up to 13.00 mm, which was significantly smaller than the 43.83 mm diameter observed for the ∆*vmsR* strain. Notably, the C∆*vmsR* strain exhibited a colony diameter averaging 13.33 mm, which is comparable to the wild-type strain GX01 (Figure 4A,B).

Bacterial swimming motility is dependent on the flagellum. To evaluate swimming motility, *Xoc* strains were inoculated in semi-solid basal medium with 0.25% agar and incubated at 28 °C for 3 days. The wild-type strain GX01 displayed a bacterial colony diameter averaging 40.17 mm, which was comparable to the 39.33 mm observed in the C∆*vmsR* strain. The ∆*vmsR* strain exhibited a significantly smaller colony diameter of 29.17 mm, as depicted in Figure 4C,D. This reduction in swimming motility in the ∆*vmsR* strain may be attributed, in part, to diminished flagellar biogenesis. Collectively, these findings indicate that VmsR plays contrasting roles in regulating swarming and swimming motility in *Xoc* GX01.

### 2.5. VmsR Specifically Binds to the fliC and fliS Promoter In Vitro and Facilitates Their Expression

The swimming motility of the ∆*vmsR* strain has decreased. Bacterial swimming motility is fundamentally reliant on the flagellum, which is a complex organelle responsible for propelling the bacterium through its environment [39]. The model of the flagellum structure is shown in Figure 5A. The RNA-Seq analysis revealed the downregulation of three genes (*fliC*, *fliD*, and *fliS*) that are associated with flagella synthesis. The mRNA expression levels of the *fliC*, *fliD* and *fliS* genes were verified by RT-qPCR. As shown in Figure 5B, the transcriptional levels of these genes were significantly reduced in the ∆*vmsR* compared to GX01, and the expression patterns of these selected genes are all consistent with that observed in the data obtained from the transcriptome analysis (Figure 3C).

To determine if the transcriptional regulation of *fliC*, *fliD*, and *fliS* is mediated by the direct binding of VmsR to the promoter region, in vitro electrophoretic mobility shift assays (EMSAs) were performed. Purified recombinant VmsR-His_6_ (Figure 5C) was incubated with 6′-carboxyfluorescein (6′-FAM)-labeled DNAs in binding buffer; then, the protein–DNA complexes were separated by electrophoresis in 4% polyacrylamide gel. The findings conclusively reveal that the recombinant VmsR-His6 exhibits a robust binding affinity to the *fliC* and *fliS* promoter regions, significantly impeding their migration within the polyacrylamide gel (Figure 5D,F). Moreover, the observed shifted bands were displaced by an excess of unlabeled probes (10×, 50×, 100×), thus confirming the specificity of VmsR’s binding to the *fliC* and *fliS* promoter. To guarantee the specificity of VmsR binding, we incorporated a negative control promoter (the *atsE* promoter), which demonstrated no affinity for VmsR. As shown in Appendix A, the EMSA results demonstrate that VmsR can directly bind to the promoter of *fliC* and *fliS* while failing to bind to the *fliD* promoter.

These results supported the hypothesis that VmsR acts as a transcriptional activator of the *fliC* and *fliS* promoter. The reduced expression of *fliD* could been indirectly regulated by VmsR.

### 2.6. EPS Production and Biofilm Formation of the ∆vmsR Strain Were Increased

In the above results, we found an interesting phenomenon. The swimming motility and swarming motility of the ∆*vmsR* strain were opposite. Liu et al. found that the self-secreted EPSs are essential for the swarming motility exhibited by *Pseudoalteromonas* sp. SM9913 [40]. Consequently, EPS was measured qualitatively and quantitatively in this study. Compared with the wild-type strain, EPS production of the ∆*vmsR* significantly increased both in qualitative and quantitative analysis. In terms of shape, the colony form of ∆*vmsR* is fuller and shinier than wild-type strain GX01 and the complementary strain C∆*vmsR* (Figure 6A). EPS production was up to 12.96 g/L for the *vmsR* mutant, significantly greater than 6.383 g/L for the wild-type strain *Xoc* GX01, but similar to 7.45 g/L for the complementary strain C∆*vmsR* (Figure 6B).

It is firmly established that the biofilm formation process of numerous bacteria necessitates the presence of both EPSs and the type IV pilus (T4P). EPSs occupy a pivotal role in this biofilm formation. EPS constitutes part of the biofilm matrix that maintains and organizes bacterial biofilms, while the T4P facilitates surface attachment as adhesins [41]. Biofilm formation was measured qualitatively and quantitatively in this study. When compared to the *Xoc* GX01 strain, the biofilm formation capacity of the ∆*vmsR* mutant was observed to be significantly elevated (Figure 6C). Quantification of the biofilm formation was achieved through the measurement of optical density at 590 nm (OD_590_). Specifically, the mean OD_590_ value for the ∆*vmsR* mutant was 3.04, which significantly surpassed the values of 1.22 recorded for *Xoc* GX01 and 1.38 for the C∆*vmsR* strain (Figure 6D).

Taken together, the results indicate that VmsR negatively affects EPS production and biofilm formation in *Xoc* GX01. Furthermore, we hypothesize that the enhanced biofilm formation and swarming motility could be attributed to the increased production of EPS.

### 2.7. VmsR Specifically Binds to the gumB Promoter In Vitro and Inhibits Its Expression

EPS is encoded by a gene cluster with *gumB* serving as one of the primary genes within this cluster. The *gumB* gene plays a pivotal role in exporting the xanthan [42]. The mRNA expression levels of *gumB* gene were verified by RT-qPCR. As shown in Figure 7A, the transcriptional levels of *gumB* gene were significantly upregulated in the ∆*vmsR* compared to GX01. To determine whether VmsR could directly regulate the *gumB* operon, in vitro EMSAs were conducted. Purified recombinant VmsR-His_6_ (Figure 5C) was incubated with a 6′-FAM-labeled DNA (440 bp) spanning from bp −455 to −16 (Appendix A) relative to the translational initiation site (TIS) of the *gumB* operon. Then, the protein–DNA complexes were separated by electrophoresis in 4% polyacrylamide gel. A distinct shift band was observed with increasing concentrations of recombinant VmsR-His_6_. Competitive binding assays with unlabeled probes at various concentrations (10×, 50×, 100×) demonstrated competitive interactions, leading to progressively fainter or absent shift bands in the presence of non-fluorescent competitive probes (Figure 7B). Taken together, these results illustrate that VmsR could specifically bind to the *gumB* operon promoter and repress the expression of *gumB* gene.

Additionally, the putative consensus sequence recognized by VmsR using the MEME Suite (https://meme-suite.org/meme/), with the promoter regions from *fliC*, *fliS* and *gumB*, identified the core DNA-binding sequence 5′-GTHGTGWWCSMWRWGKYTT-3′ (H: A/C/T; W: A/T; S: C/G; M: A/C; R: A/G; K: G/T; Y: C/T) (Appendix A).

## 3. Discussion

In this work, we identified a LuxR-type transcriptional regulator VmsR, which acts as a global regulator influencing diverse cellular processes including cell motility, EPS production and biofilm formation in *Xoc* GX01. Specifically, we have identified VmsR as a key regulator of flagellum in *Xoc*. Our findings indicate that VmsR directly promotes the expression of *fliC* and *fliS* genes, revealing its crucial role in positively regulating motility and pathogenicity in *Xoc*.

LuxR-type regulators are known to play a role not only in the QS system but also in other biological functions such as flagellar synthesis. For example, AcrR, a LuxR-type regulator, regulates flagellar assembly and contributes to the virulence, motility, biofilm formation, and growth ability of *Acidovorax citrulli* [38]. VisN and VisR, two LuxR-type regulators in *Sinorhizobium meliloti*, act as global regulators of chemotaxis, flagellar, and motility genes [43]. AclR, also a LuxR-type global regular, regulates the motility and virulence of *A. citrulli* [44]. In accordance with the aforementioned LuxR-type regulators, VmsR similarly performs a crucial function in governing chemotaxis, flagellar assembly, and motility-associated genes. As demonstrated in our study, VmsR is a TCS response regulator, not a traditional QS system regulator. The *vmsR* mutant strain exhibits a reduced swimming motility but an increased swarming motility, EPS production, and biofilm production. This demonstrated that VmsR either positively or negatively regulated multiple biological functions of *Xoc* GX01.

The flagellum is involved not only in the movement and chemotaxis of bacteria but also in several functions associated with bacterial pathogenicity, including biofilm formation, protein export, and adhesion [45]. The flagellum also serves as a virulence factor in many bacterial pathogens. Steffens et al. found that the *fliC* mutant inhibited the motility and showed an increased xanthan production in *Xcc* JBL007 [46]. In line with the aforementioned findings, the deletion of the *vmsR* gene in *Xoc* GX01 resulted in a significantly reduced expression of *fliC*, which was accompanied by an enhanced EPS production. In our study, the deletion of the *vmsR* gene led to a diminished expression of the *fliC* gene, potentially causing a reduction in swimming motility, which contributed significantly to the decreased virulence exhibited by the bacterium on rice. Our RNA-Seq data support the role that VmsR plays in flagellar biosynthesis because three genes involved in flagellar assembly were differentially expressed in the *vmsR* mutant strain in comparison with the wild-type strain.

In this study, ∆*vmsR* exhibited a decrease in swimming motility and an increase in swarming motility. The result shows that VmsR plays an important role in the cell motility of *Xoc* GX01. Swimming motility is an individual movement in liquid or semi-solid media, and it requires flagella, which is a complex organelle responsible for propelling the bacterium through its environment [47]. Song et al. discovered that flagellar instability results in a diminished virulence of *Aeromonas veronii* and correlates with its pathogenicity [48]. In this study, three flagellar genes, namely *fliC*, *fliD*, and *fliS*, were downregulated through the RT-qPCR analysis. In vitro EMSA assay, VmsR can directly bind to the promoter of *fliC* and *fliS*. *fliD* encodes the HAP2/filament cap protein, which plays a pivotal role in regulating the assembly and stability of bacterial flagella. *fliC* encodes the H1 flagellin protein, which is a key component of the flagellar filament responsible for motility. FliS functions as a specific export chaperone for FliC, facilitating its translocation and assembly within the bacterial cell [49,50]. In general, the expression of these genes is negatively regulated by the secretion of FlgM, which is a regulatory protein that modulates flagellar biosynthesis and function [51]. FliS, on the other hand, operates as a non-canonical chaperone, fine-tuning FlgM’s activity to regulate the expression of late flagellar genes as well as motility and biofilm formation in *Yersinia pseudotuberculosis* [51]. Therefore, we speculate that VmsR has positively regulated the expression of *fliC* and *fliS*. The deletion of the *vmsR* gene results in the instability of flagella, causing a decrease in the swimming motility. These results show that the flagellum is very important for pathogenic bacteria. Consequently, the reduction pathogenicity of ∆*vmsR* may be caused by reduced flagellar synthesis.

Swarming motility is a group movement on semi-solid medium, which requires both flagella and type Ⅳ pili [52]. Swarming motility could increase the expression of virulence genes such as type Ⅲ effector and the type Ⅱ secretory system [53,54]. Swarming motility, regulated by several genes and pathways like the QS system, is a complex bacterial behavior [55]. One study showed that swarming is positively regulated by rhamnolipid production [56]. As in other *Xanthomonas* [57], the difference of swimming motility and swarming motility remains unknown, requiring further research. Nevertheless, previous research has shown that the self-secreted EPS is positively correlated to the swarming behavior of *Pseudoalteromonas* sp. SM9913 [40]. Consequently, the enhanced EPS production in the ∆*vmsR* mutant may offer a plausible explanation for the remarkable swarming motility that has been observed.

Biofilm is a critical virulence factor for many plant pathogens [58,59]. When *Xanthomonas citri* was impaired in its ability to form a biofilm, it exhibited mitigated disease symptoms [60]. Biofilm formation involves some adhesive molecules, such as polysaccharide or protein. EPS usually contributes to biofilm formation in *Xoc* [61]. In general, the enhanced biofilm provides a protective barrier that could contribute to increased virulence by facilitating the colonization of host tissues and evasion of host immune responses. The mutant of *rbdA* in *Pseudomonas aeruginosa* PAO1 results in increased EPS production and biofilm formation [62]. *P. aeruginosa* PAO1 has two operons, *pel* and *psl*, that have been reported to contribute to EPS production [63]. They proved that the increased expression of the *pel* operon might contribute to the hyperbiofilm phenotype of the Δ*rbdA* mutant [62]. Likewise, the mutation of *bifA* in P. aeruginosa elicits an enhanced biofilm phenotype. Furthermore, BifA has been demonstrated to regulate biofilm formation through modulating the expression of the *pel* operon [64]. Consistent with the above studies, the EPS production and biofilm formation in the Δ*vmsR* mutant were increased, which are hypothesized to confer significant advantages to its pathogenicity. Nevertheless, the pathogenicity of the Δ*vmsR* mutant has decreased. We presume that EPS, being a co-pathogenic factor, does not necessarily exhibit a direct and positive correlation with pathogenicity. High levels of EPS production significantly contribute to the robustness of the biofilm structure, thereby potentially shielding the bacteria from host immune responses [61]. However, this augmentation in robustness also diminishes the bacteria’s capacity to disperse, potentially reducing its virulence in dynamic environments. From an ecological long-term perspective, bacteria excrete substantial quantities of EPS as a strategic defense mechanism to evade the host’s immune system. This temporary state of defense, rather than aggression, facilitates the enhanced survival of the bacterial colony as a whole.

Additionally, flagella are suggested to help *E. coli* overcome repulsive surface forces to establish initial contact [65]. Flagella contribute to the early stages of biofilm formation in Bacillus cereus by facilitating surface attachment and microcolony formation [66]. However, when *E. coli* express other surface structures like Curli or conjugative pili, flagella are not necessary for adhesion or biofilm development [67,68]. This suggests that flagellar motility alone is not the key factor for promoting adhesion. On the other hand, flagella may serve a mechanosensory role in detecting surfaces, which triggers a transition from planktonic (free-swimming) to sessile (adherent) states, which are essential for biofilm formation [69,70]. When surfaces are detected, flagella-mediated motility is downregulated, possibly due to the influence of the cellular messenger c-di-GMP, which is linked to both the suppression of motility and the promotion of robust biofilm formation [71,72]. This dual role of flagella—both promoting initial surface contact and then downregulating motility upon surface detection—underscores a complex regulation where motility and biofilm formation are generally mutually exclusive. In this work, biofilm formation of the ∆*vmsR* strain was increased significantly compared with the wild-type strain GX01, while the swimming of the ∆*vmsR* strain was reduced. These results are consistent with the above studies. Motility and biofilm are mutually exclusive lifestyles, and shifts between the two are under the strict regulation of bacteria attempting to adapt to the complexity and variability of environmental conditions.

Unexpectedly, EPS production has risen in ∆*vmsR*, yet the *gum* gene is conspicuously absent from the DEGs. Similarly, this phenomenon aligns with the observation that the deletion of *vemR* conspicuously diminished EPS production despite the absence of any *gum*-related genes in the DEGs [73]. In *Xoc*, the *gum* cluster, encompassing 13 genes ranging from *gumB* to *gumN*, is vital for EPS assembly, polymerization, and export [74,75]. It has been demonstrated that in *Xcc*, EPS synthesis is initiated in the late-exponential growth phase and reaches maximal production during the stationary growth phase, and the expression of the *gum* genes mirrors the time course of EPS production [76]. As described above, our transcriptome analysis used the bacterial cells in the mid-exponential growth phase. Therefore, this inconsistency is probably due to the employment of bacterial cells from different growth phases in the experiments.

Given their central role in bacterial physiology and pathogenesis, HKs represent promising targets for the development of novel antimicrobial agents [77]. While the cognate HKs of VmsR remains unknown, further research efforts are likely to shed light on this important aspect of bacterial signaling. In addition, RR proteins can also be used as direct targets. Identifying the sensor kinase responsible for activating VmsR will not only deepen our understanding of TCS function but also have implications for the development of novel antibacterial strategies and biotechnology applications.

In summary, the *vmsR* gene occupies a crucial position in the virulence of *Xoc* GX01, modulating swimming motility and flagellar formation positively, while suppressing EPS production and biofilm formation, either directly or indirectly. Transcriptomic investigations have further illuminated that VmsR adversely affects methyl-accepting chemotaxis proteins, which are integral to motility. The presence of a response regulatory domain at the N-terminus of VmsR suggests its potential to interact with a diverse array of signals regulating biological functions within *Xoc* GX01. However, the precise nature of these signals remains an enigma. Future research is imperative to decipher these signals and elucidate the underlying molecular mechanisms governing VmsR’s regulation in *Xoc* GX01, ultimately aiming to devise efficient control strategies against this significant bacterial pathogen.

## 4. Materials and Methods

### 4.1. Bacterial Strains, Culture Media and Growth Conditions

*Xoc* strains were routinely grown in nutrient broth (NB) medium (beef extract, 3 g/L; yeast extract, 1 g/L; polypeptone, 5 g/L) or on NB agar plates at 28 °C. *E. coli* strains were cultivated in Luria–Bertani (LB) medium or on LB agar plates at 37 °C. When necessary, antibiotics were added to medium at the following concentrations: kanamycin, 25 μg/mL; rifampicin, 50 μg/mL. The *Xoc* strains and plasmids used in this work are listed in Appendix A.

### 4.2. Construction of the vmsR Deletion Mutant and Its Complementary Strain

The *vmsR* gene was knocked out in *Xoc* GX01 using a homologous recombination method. The follow primers used in this study are listed in Appendix A. The 492 bp upstream and 522 bp downstream sequences of the *vmsR* gene were amplified from the *Xoc* wild-type strain GX01 genome using the *vmsR*-LF/LR and *vmsR*-RF/RR primers. The amplified DNA fragments were digested by corresponding restriction enzymes and cloned into pK18*mobsacB*, which is a conjugative suicide plasmid in *Xoc*. Then, the resulting recombinant plasmids were created in *E. coli* strain DH5α and transferred to wild-type strain *Xoc* GX01 by electroporation (2 mm, 2.5 kV). Transconjugants were screened on NA medium with 10% sucrose and antibiotics (Rif^r^ and Kam^r^). Transconjugants were identified by external primers (*vmsR*-LF and *vmsR*-RR) and *vmsR* ORF internal primers (*vmsR*-inF and *vmsR*-inR). The successful transconjugants should satisfy the following two points at the same time: the combined fragment of upstream and downstream sequences was amplified by external primers, and no PCR product could be amplified by internal primers.

To complement the deletion mutant, the 989 bp sequence of the *vmsR* gene was amplified using the *vmsR*-cmF and *vmsR*-cmR primers. The PCR fragments were digested with the *Xba*Ⅰ/*Hin*dⅢ restriction enzymes and ligated into *Xba*Ⅰ/*Hin*dⅢ-digested pXUK, resulting in the recombinant plasmids. pXUK was derived from the endogenous plasmid pXOCgx01 isolated from *Xoc* GX01 [78]. Recombinant plasmids were created in *E. coli* strain DH5α and transferred to the *vmsR* mutant of *Xoc* by electroporation (2 mm, 2.5 kV). Transconjugants were screened on NA medium with Rif^r^ and Km^r^ antibiotics. Transconjugants were identified by *vmsR*-cmF/R primers.

### 4.3. Virulence Assay

The virulence of *Xoc* was determined by the pressure inoculation of leaves in rice [79]. *Xoc* strains were grown in NB medium at 28 °C with shaking at 200 rpm until the value of OD_600_ reached 0.8–1.0. The value of OD_600_ was adapted to 0.5 using sterilized ddH_2_O. The above bacterial cell suspensions were inoculated into the leaves of 6-week-old rice plants using a needleless injector. Water-soaking symptoms were measured 15 days after inoculation. We also recorded the lesion length on at least 30 leaves.

### 4.4. Assay for Swimming and Swarming Motilities

To determine the swimming and swarming motilities of the *vmsR* mutant strain and the wild-type strain *Xoc* GX01, *Xoc* strains were cultured in NB medium at 28 °C with shaking at 200 rpm until the value of OD_600_ reached 1.0.

For swimming motility, the value of OD_600_ was adjusted to 1.0. A volume of 3 μL of each bacterial cell suspension was injected into the center of a basal medium plate containing 0.3% agar and then incubated at 28 °C for 3 days. For swarming motility, the value of OD_600_ was adjusted to 0.2. A volume of 2 μL of each bacterial cell suspension was precisely dispensed onto the center of an NB medium plate containing 2% sucrose and 0.6% agar and then incubated at 28 °C for 3 days. Due to the low strength of the medium plate, the glass garden should be placed upward. After 3 days of incubation at 28 °C, the diameter of the bacterial colony on each plate was determined. In both swimming and swarming motility assays, each treatment involves three replications, and each experiment was repeated three times.

### 4.5. Determination of Biofilm Formation and EPS Production

Biofilm formation on glass surfaces was performed as described previously [17]. Briefly, *Xoc* strains were grown in NB medium at 28 °C with shaking at 200 rpm until the value of OD_600_ reached 1.0. The value of OD_600_ was adapted to 0.5, and then 1.0 mL of cultures prepared as aforementioned was added in borosilicate glass tubes (100 mm * 15 mm). These cultures were incubated at 28 °C for 5 days without shaking. We removed the cell suspension from the borosilicate glass tubes and washed the glass tubes with sterile water three times. Then, 2.0 mL of 0.1% (*w*/*v*) crystal violet was added to these glass tubes for 30 min. To quantify the biofilm formation, stained biofilm solubilized with 2 mL absolute ethanol was measured with a spectrophotometer under OD_600_. Each treatment involved three replications, and the same experiment was repeated three times.

For analysis of EPS production on plates, the *Xoc* strains were cultured in NB medium at 28 °C with shaking at 200 rpm until the value of OD_600_ reached 0.8–1.0. The value of OD_600_ was adjusted to 0.2, and then 2 μL of each strain was inoculated onto the surface of NA plates containing 2% sucrose. After 48~72 h of incubation at 28 °C, the colony diameter of the *Xoc* strains was measured. To estimate EPS production, the *Xoc* strains were inoculated into 100 mL of NB liquid medium containing sucrose (2% *w*/*v*) at 28 °C, 200 rpm for 5 days. EPS was precipitated from the culture supernatant with ethanol, dried at 55 °C and weighed as described. The results were observed after 5 days of incubation at 28 °C. Each treatment involves three replications, and the same experiment was repeated three times.

### 4.6. Transcriptome Analysis

*Xoc* strains were cultured in NB medium at 28 °C with shaking at 200 rpm until the OD_600_ value reached 1.0. Three biological replicates were derived from the bacteria solution. The bacterial cells were collected using high-speed centrifuge and then frozen with liquid nitrogen. The bacteria cells were sent to the PFOMIC Bioinformatics Company (Nanning, China) for library preparation and strand-specific transcriptome sequencing. Sequencing libraries were generated using the NEBNext^®^ Ultra^TM^ Directional RNA Library Prep Kit for Illumina^®^ (NEB, USA) following the manufacture’s recommendations, and index codes were added to attribute sequences to each sample. HTSeq v0.6.1 was used to count the read numbers mapped to each gene. And then, the FPKM of each gene was calculated based on the length of the gene and the reads count mapped to this gene. FPKM, the expected number of Fragments Per Kilobase of transcript sequence per Millions base pairs sequenced, considers the effect of sequencing depth and gene length for the reads count at the same time, and it is currently the most commonly used method for estimating gene expression levels [80]. Differential expression analysis of two conditions was performed using the DEGSeq R package (1.20.0). The *p* values were adjusted using the Benjamini and Hochberg method [81]. A corrected FDR of 0.05 and log_2_(Fold Change) of 1 were set as the threshold for significantly differential expression.

### 4.7. RNA Isolation and Quantitative Reverse Transcription PCR Analysis of Gene Expression

*Xoc* strains were grown in NB medium at 28 °C with shaking at 200 rpm until the value of OD_600_ reached 1.0. The bacterial cells were collected using high-speed centrifuge and then washed three times with sterilized water. RNA was isolated from the *Xoc* strains using a TransZol Up Plus RNA Kit (TransGen, Beijing, China). To confirm the reality of the RNA-seq data, two-step quantitative reverse transcription PCR (RT-qPCR) was adapted in this work. First-strand cDNA was synthesized using RT SuperMix for qPCR (Vazyme Biotech, Nanjing, China). The cDNA was added as DNA templates into the PCR system with the 2 × RealStar Green Fast Mixture (GenStar, Beijing, China). The qPCR adapted a three-step amplification procedure. Firstly, the template cDNA was pre-denatured at 95 °C for 2 min. A total of 40 cycles were amplified by qPCR, and each cycle was performed using the following procedure: 95 °C for 15 s, 60 °C for 15 s, and 72 °C for 30 s. The 16S rRNA gene of *X. oryzae* pv. *oryzicola* was used as the internal control to verify the absence of significant variation at the cDNA level in the samples.

### 4.8. In Vitro Electrophoretic Mobility Shift Assay (EMSA)

The EMSA represents a swift and precise technique for detecting interactions between proteins and nucleic acids. The principle is that the electrophoretic mobility of a protein–DNA complex is typically less than that of the free DNA [82,83]. DNA probes were amplified by PCR with the corresponding primers (Appendix A). The 5′-end of the reverse primer was modified with 6′-FAM (Sangon Biotech, Shanghai, China). DNA fragments encoding the putative DNA-binding domain of VmsR were cloned into the plasmid pRSFDuet-1 to generate the recombinant plasmids. The recombinant plasmids were introduced into fresh competent cells of *E. coli* BL21 (DE3) Chemically Competent Cell (CD601, TransGen, Beijing, China). The recombinant plasmids were incubated in Luria–Bertani medium for 3 to 5 h at 37 °C with a shaking speed of 200 rpm, achieving an OD_600_ of 0.6 to 0.8. Subsequently, 0.5 mM isopropyl-β-d-thiogalactoside (IPTG) (Solarbio Life Sciences, Beijing, China) was added, and the culture was further maintained for 12 h at 16 °C with a shaking speed of 120 rpm. *E. coli* BL21 (DE3) cells were collected and disrupted to extract the recombinant polypeptides. The recombinant proteins were purified by ProteinIso^®^ Ni-NTA Resin (TransGen, Beijing, China). The DNA probe (40 ng) was mixed with different amounts of recombinant protein (0–3 μg) in binding buffer (1 mM dithiothreitol, 0.1 mg/mL BSA, 50 mM KCl, 20 mM Tris-HCl, pH 8.0, 5% glycerol) and then reacted at 25 °C for 30 min. The protein–DNA complexes were separated by electrophoresis in 4% polyacrylamide gel (acrylamide/bisacrylamide, 29:1) in 0.5 × Tris-borate-EDTA (TBE) buffer (44.5 mM Tris base, 44.5 mM boric acid, and 1 mM EDTA, pH 8.0) and recorded with a Bio-Rad ChemiDoc™ MP Imaging System (Bio-Rad Laboratories, Hercules, CA, USA). For the competitive EMSA assay, DNA probes without 6′-FAM were substituted.

As shown in Appendix A, the *fliC* probe fragment measures 283 bp, encompassing the region from base pair −245 to +38, relative to the TIS of the *fliC* promoter. The *fliD* probe fragment spans 272 bp, extending from base pair −272 to −1 with respect to the TIS of the *fliD* promoter. Similarly, the *fliS* probe fragment encompasses 500 bp, covering the sequence from base pair −500 to −1, relative to the TIS of the *fliS* promoter.

## 5. Patents

China Invention Patent: Authorization Bulletin Number: CN112852839B (Published: 2 September 2022).

## Figures and Tables

**Figure 1 ijms-25-07595-f001:**
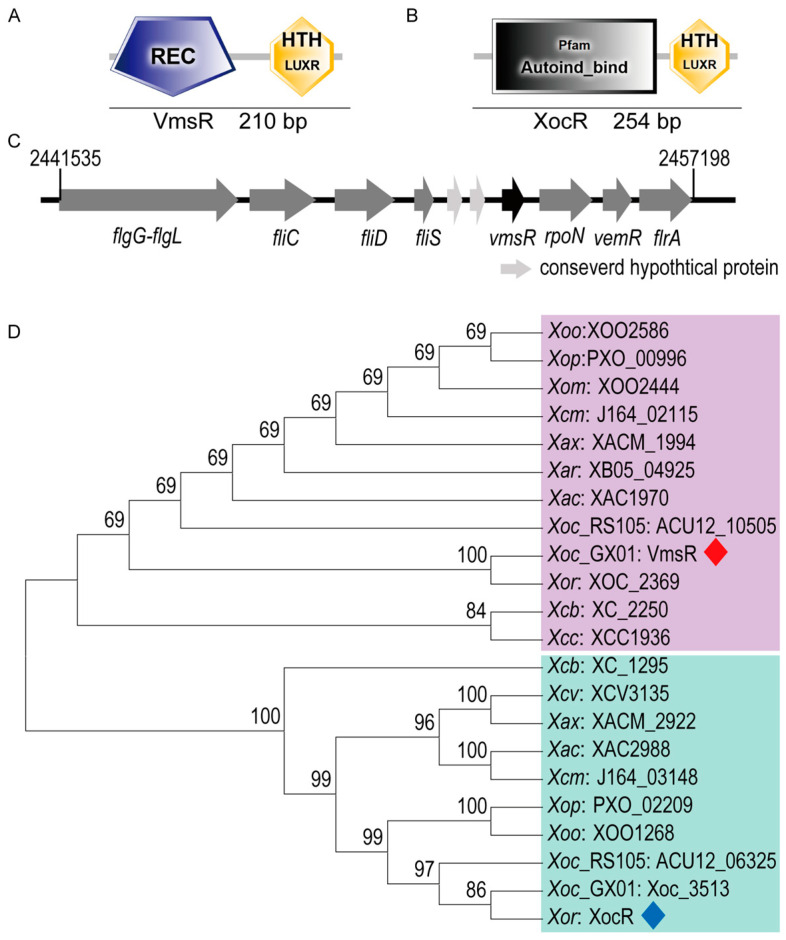
Gene arrangement and phylogenetic trees of VmsR ortholog proteins from *Xanthomonas* species. (**A**) Domain structure of VmsR. REC, response regulator domain; HTH LuxR, LuxR family helix–turn–helix domain. (**B**) Domain structure of XocR. Autoind_bind domain; HTH LuxR, LuxR family helix–turn–helix domain. (**C**) Arrangement of *vmsR* and surrounding genes. (**D**) Phylogenetic trees constructed from amino acid sequence alignments of VmsR ortholog proteins from *Xanthomonas* species. To highlight VmsR and XocR, VmsR was remarked with a red rhombus, while XocR was distinguished with a blue rhombus. The evolutionary history was inferred using the Neighbor-Joining method. The percentage of replicate trees in which the associated taxa clustered together in the bootstrap test (1000 replicates) are shown next to the branches. The tree is drawn to scale, with branch lengths in the same units as those of the evolutionary distances used to infer the phylogenetic tree. The evolutionary distances were computed using the *p*-distance method and are in the units of the number of amino acid substitutions per site.

**Figure 2 ijms-25-07595-f002:**
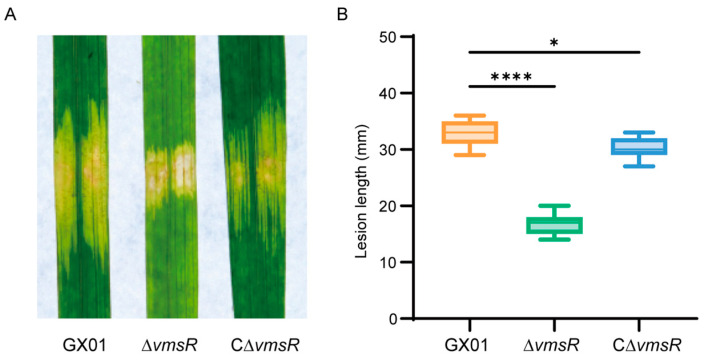
Virulence of the ∆*vmsR* was decreased. (**A**) Fifteen days after inoculation, the lesion length of the wild-type strain GX01, ∆*vmsR* and complementary strain C∆*vmsR* were recorded. (**B**) The virulence of *Xoc* was impaired when *vmsR* gene was deleted. Data are displayed as boxplots with individual data points. Error bars represent maximum and minimum values. Values given are the means ± *SD* of fifteen measurements from a representative experiment. Vertical bars represent standard errors (* *p* < 0.05, **** *p* < 0.0001, *t* = test).

**Figure 3 ijms-25-07595-f003:**
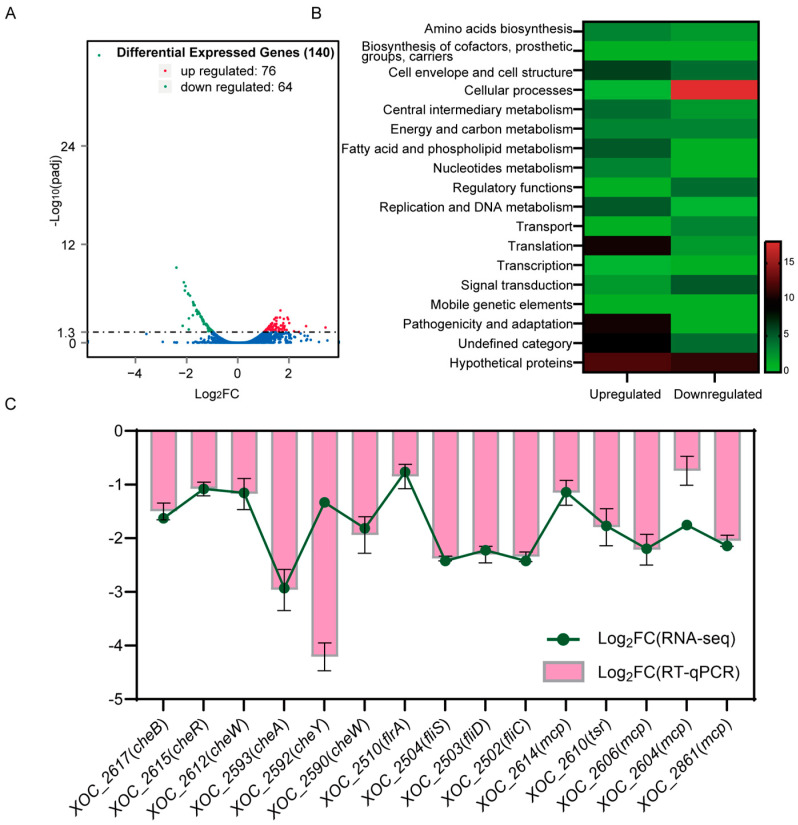
RNA sequencing and functional classification of differentially expressed genes. (**A**) Identified DEGs between ∆*vmsR* and GX01. Volcano plots showing the expression level of each unigene. Limits defined by FDR ≤ 0.05 and |Fold Change| ≥ 2. Red dots represent upregulated genes in ∆*vmsR*, green dots represent downregulated genes in ∆*vmsR*, and blue dots represent statistically unchanged genes. (**B**) Functional classification of the 140 DEGs in the ∆*vmsR* of *Xoc* GX01. The transcriptomes of *Xoc* strains cultured in NB medium were investigated by RNA−Seq. Among the 140 DEGs, 76 genes and 64 genes were upregulated and downregulated, respectively. These genes were broadly categorized according to their biological function. The DEGs list is shown in Appendix A. (**C**) The transcription level of several motility−related genes in ∆*vmsR* was estimated by quantitative reverse transcription PCR (RT−qPCR). *Xoc* strains were grown in NB medium to a concentration of OD_600_ of 0.8−1.0, and RNA was isolated. Relative gene expression with respect to the corresponding transcript levels in the wild−type strain GX01 was calculated. Values given are the means ± *SD* of triplicate measurements from a representative experiment. Vertical bars represent standard errors (*p* < 0.01; *t* = test). Similar results were obtained in three other independent experiments.

**Figure 4 ijms-25-07595-f004:**
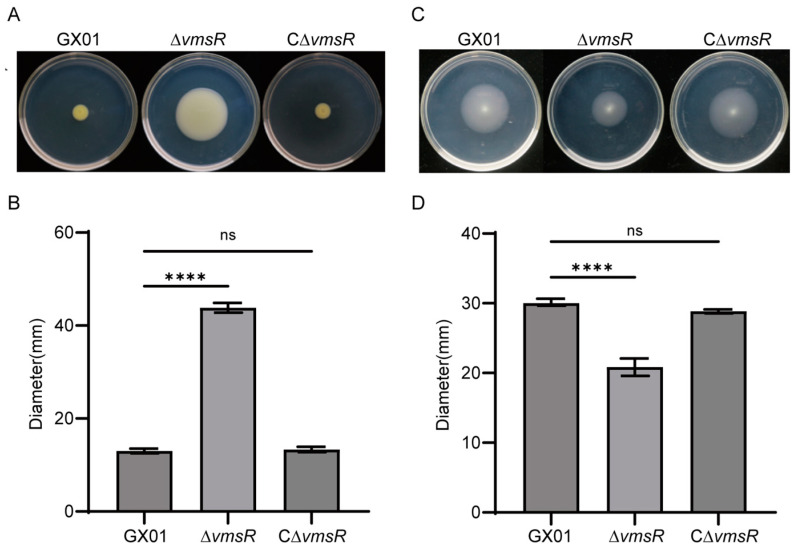
Compared to *Xoc* GX01, the ∆*vmsR* exhibited an enhanced swarming motility but a reduced swimming motility. (**A**,**B**) The swarming motility of the ∆*vmsR* strain has increased. Measured data of diameters of the swarming motility on the plate of semi-solid NA medium with 0.6% agar. Each treatment involves three replications, and each experiment was repeated three times. Vertical bars represent standard errors (ns > 0.05, **** *p* < 0.0001, *t* = test). (**C**,**D**) The swimming motility of the ∆*vmsR* strain has decreased. Measured data of diameters of the swimming motility on the plate of basal medium with 0.25% agar. Each treatment involves three replications, and each experiment was repeated three times. Vertical bars represent standard errors (ns > 0.05, **** *p* < 0.0001, *t* = test).

**Figure 5 ijms-25-07595-f005:**
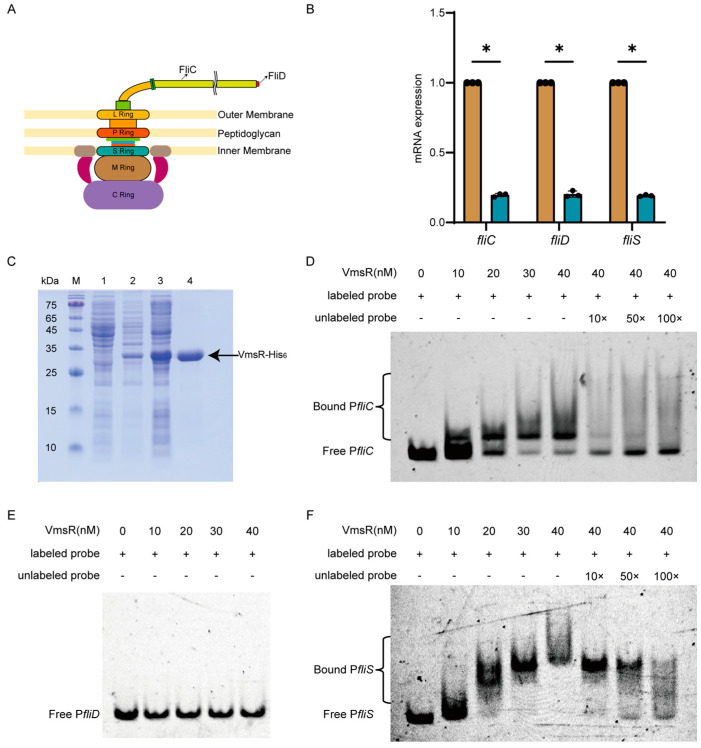
VmsR specifically binds to the *fliC* and *fliS* promoter in vitro and facilitates their expression. (**A**) The diagram of flagellar pattern. (**B**) The transcription level of *fliC, fliD* and *fliS* genes in the ∆*vmsR* strain was estimated by RT−qPCR. The asterisks indicate significant differences compared with the wild−type GX01 strain (mean ± *SD*, *n* = 3, * *p* < 0.05). (**C**) 12.5% SDS−PAGE gel plots of VmsR−His_6_ protein purification. M: StarRuler Color Prestained Protein Marker (10−180 kDa) (M221, GenStar, Beijing, China). lane1: cytosol in the absence of IPTG induction. lane2: the precipitates after IPTG induction. lane3: the supernatant after IPTG induction. lane4: purified VmsR−His_6_ protein. (**D**–**F**) In vitro electrophoretic mobility shift assay (EMSA) between the recombinant protein VmsR−His_6_ and the promoter regions of *fliC, fliD* and *fliS* genes. Each reaction contained VmsR (10−40 nM) and 6′−FAM-labeled probes (~10 ng).

**Figure 6 ijms-25-07595-f006:**
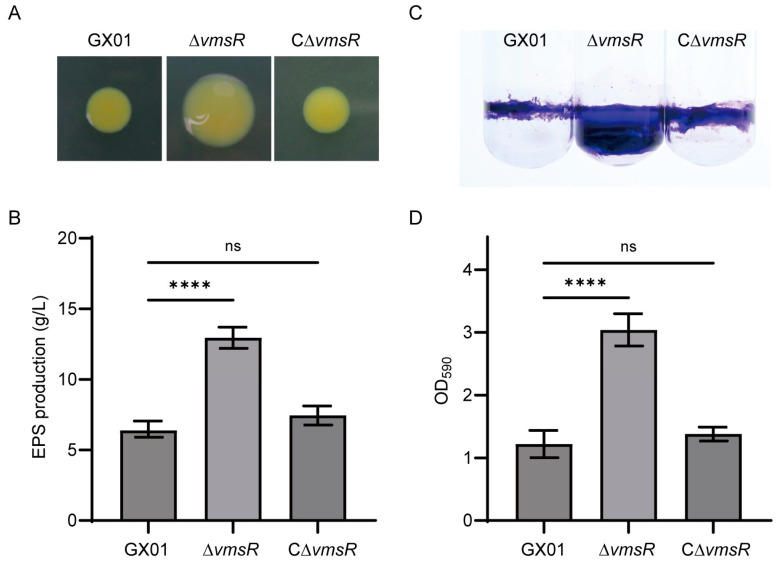
EPS production and biofilm formation of the ∆*vmsR* mutant has increased compared with wild-type strain GX01. (**A**,**B**) The EPS production was qualitatively and quantitatively assessed in the wild-type strain GX01, the ∆*vmsR* mutant, and the complemented strain C∆*vmsR*. Additionally, the dry weight of each *Xoc* strain was measured. Vertical bars represent standard errors (ns > 0.05, **** *p* < 0.0001, *t* = test). Similar results were obtained in three other independent experiments. (**C**,**D**) Biofilm of wild-type strain GX01, ∆*vmsR* and C∆*vmsR* was stained with 0.1% (*w*/*v*) crystal violet. Quantification of biofilm formed by wild-type strain GX01, ∆*vmsR* and C∆*vmsR.* Biofilm formation measured under OD_590_ using stained biofilm solubilized with absolute ethanol. Vertical bars represent standard errors (ns > 0.05, **** *p* < 0.0001, *t* = test).

**Figure 7 ijms-25-07595-f007:**
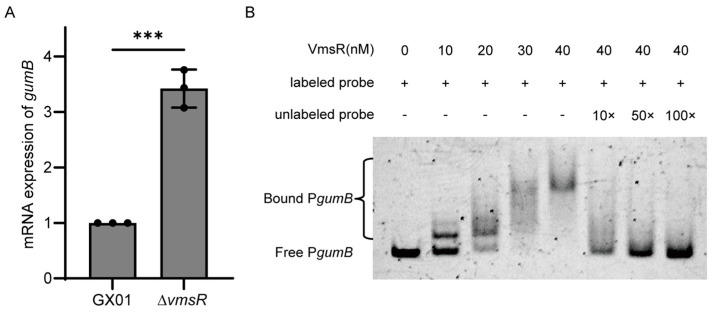
VmsR specifically binds to the *gumB* promoter in vitro and inhibits its expression. (**A**) The transcription level of *gumB* in ∆*vmsR* was estimated by RT−qPCR. The asterisks indicate significant differences compared with the wild−type *Xoc* GX01 (mean ± *SD*, *n* = 3, *** *p* < 0.001). (**B**) In vitro electrophoretic mobility shift assay (EMSA) between the recombinant protein VmsR and the promoter regions of *gumB* genes. Each reaction contained VmsR (10−40 nM) and 6′-FAM-labeled probes (~10 ng).

## Data Availability

The RNA sequencing data generated in this study are available in the NCBI SRA database under the accession codes SRR28710641-SRR28710646 (https://www.ncbi.nlm.nih.gov/sra/PRJNA1101110, accessed on 16 April 2024). Other data are presented within the manuscript and Appendix A.

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
