# Peer review of "VmsR, a LuxR-Type Regulator, Contributes to Virulence, Cell Motility, Extracellular Polysaccharide Production and Biofilm Formation in Xanthomonas oryzae pv. oryzicola"

_ijms, 2024, doi:10.3390/ijms25147595_

Round 1

Reviewer 1 Report

Comments and Suggestions for Authors

I congratulate the authors for their work and manuscript. Manuscript ID: ijms-3093529 "VmsR, a LuxR-type regulator, contributes to virulence, cell motility, extracellular polysaccharide production and biofilm formation in Xanthomonas oryzae pv. oryzicola" by Zhang and collaborators is well written, with methods clearly described including statistical analyses. The Discussion and conclusions are supported by the results, with little speculation. The manuscript shows a detailed characterization of a novel transcriptional regulator VmsR, involved in various important cellular behaviors of this aggressive tice pathogen. The authors successfully obtained and compared mutant and complemented strains, and analyzed the effects on gene expression by RNA-seq and RT-qPCR. Bacterial motility and exopolysaccharide production were also measured, as well as EMSA to demonstrate interaction with DNA regulatory sequences. Taken together, the results provide a deep characterization of this protein. Perhaps one additional analysis that could be included is a bioinformatic search for regulatory motifs in the promoter regions of the genes analyzed by EMSA to depict putative consensus sequence(s) recognized by VmsR. I'll recommend acceptance after a minor review to give the authors the opportunity to work on this. Besides this suggestion, I have a few other minor suggestions that I listed below and can easily be implemented in the final version.

Line 13: To avoid repeating "bacterial" in the same sentence, I recommend substituting "a bacterial pathogen of rice" for "a rice pathogen".
Line 20: review "filS". Did the authors mean "fliS"?
Line 31: "Xoc is a Gram-negative bacterium..."
Line 59-60: Please review the sentence for clarity. Did the authors mean "...both a receptor and a response domain."?
Line 397: Please review the sentence for clarity.
Line 537: "The bacterial cells...".

Comments on the Quality of English Language

English is overall fine but minor improvements could be made by the editing team.

Reviewer 2 Report

Comments and Suggestions for Authors

Review comment on “VmsR, a LuxR-type regulator, contributes to virulence, cell motility, extracellular polysaccharide production and biofilm formation in Xanthomonas oryzae pv. oryzicola”

General comment:

This study investigates the role of VmsR, a LuxR-type regulator, in Xanthomonas oryzae pv. oryzicola (Xoc), focusing on its impact on virulence, cell motility, extracellular polysaccharide (EPS) production, and biofilm formation. The research provides valuable insights into the regulatory mechanisms of this important rice pathogen. The authors employ a range of molecular and phenotypic techniques to characterize the function of VmsR, presenting evidence for its role in regulating key virulence factors. While the study offers interesting findings, there are areas where additional experiments or clarifications could strengthen the conclusions.

Specific comments:

1.       The authors should provide more background on the significance of Xanthomonas oryzae pv. oryzicola in rice cultivation to contextualize the importance of this research for a broader audience.

2.       The phylogenetic analysis of LuxR-type proteins (Figure 1C) could benefit from a more detailed explanation of its implications for VmsR function in Xoc.

3.       The virulence assay results (Figure 2) are interesting, but the authors should consider including additional time points to show the progression of infection.

4.       The RNA-seq analysis provides valuable information, but it would be helpful to see a more in-depth discussion of the most significantly affected pathways or gene clusters.

5.       The contrasting effects of VmsR deletion on swimming and swarming motility are intriguing. The authors should elaborate on potential mechanisms underlying this difference.

6.       The EMSA results (Figure 5) provide evidence for direct binding of VmsR to certain promoters. It would be beneficial to include negative control promoters to further demonstrate binding specificity.

7.       The authors should discuss the potential limitations of using a single mutant strain for their analyses and consider creating additional mutants or complementation strains to strengthen their conclusions.

8.       The increased EPS production and biofilm formation in the ΔvmsR mutant is interesting. The authors could expand on the potential ecological or pathological implications of these phenotypes.

9.       The discussion could be enhanced by comparing the functions of VmsR to similar regulators in other Xanthomonas species or related plant pathogens.

10.    Given the importance of VmsR in regulating virulence factors, the authors should discuss potential applications of this research for developing new strategies to control bacterial leaf streak in rice.
